# Towards Transparent Healthcare: Advancing Local Explanation Methods in Explainable Artificial Intelligence

**DOI:** 10.3390/bioengineering11040369

**Published:** 2024-04-12

**Authors:** Carlo Metta, Andrea Beretta, Roberto Pellungrini, Salvatore Rinzivillo, Fosca Giannotti

**Affiliations:** 1Institute of Information Science and Technologies (ISTI-CNR), Via Moruzzi 1, 56127 Pisa, Italy; andre.beretta@isti.cnr.it (A.B.); rinzivillo@isti.cnr.it (S.R.); 2Faculty of Sciences, Scuola Normale Superiore, P.za dei Cavalieri 7, 56126 Pisa, Italy; roberto.pellungrini@sns.it (R.P.); fosca.giannotti@sns.it (F.G.)

**Keywords:** artificial intelligence, explainable artificial intelligence, machine learning

## Abstract

This paper focuses on the use of local Explainable Artificial Intelligence (XAI) methods, particularly the Local Rule-Based Explanations (LORE) technique, within healthcare and medical settings. It emphasizes the critical role of interpretability and transparency in AI systems for diagnosing diseases, predicting patient outcomes, and creating personalized treatment plans. While acknowledging the complexities and inherent trade-offs between interpretability and model performance, our work underscores the significance of local XAI methods in enhancing decision-making processes in healthcare. By providing granular, case-specific insights, local XAI methods like LORE enhance physicians’ and patients’ understanding of machine learning models and their outcome. Our paper reviews significant contributions to local XAI in healthcare, highlighting its potential to improve clinical decision making, ensure fairness, and comply with regulatory standards.

## 1. Introduction

The advent of artificial intelligence (AI) technologies has been transformative for healthcare, offering unprecedented capabilities in disease diagnosis, patient outcome prediction, and the development of tailored treatment plans. However, the increasing complexity and opacity of AI algorithms have amplified the need for transparency and interpretability in their decision-making processes. This has led to the rise of Explainable Artificial Intelligence (XAI), particularly focusing on local interpretation methods such as the LORE (Local Rule-Based Explanations) [1], to make AI decisions in healthcare settings more transparent and comprehensible [2,3].

XAI seeks to equip AI algorithms with explanations, enabling healthcare professionals to gain insights into the rationale behind AI-generated decisions and predictions. XAI methodologies can be broadly divided in two families: *global* [4] and *local* [5] methods. While global methods aim at explaining the general reasoning of an AI model, local methods have the goal of explaining why an AI model gave a certain output for a particular instance, i.e., the data of a particular patient or the diagnostic images belonging to a specific individual. The application of local XAI methods addresses the demand for precise, case-by-case explanations, which are paramount for clinical decision making, enhancing patient care, and fostering trust in AI systems among healthcare providers and patients [6,7,8,9,10,11,12,13].

**Significance of Local XAI in Healthcare**. The integration of AI in healthcare has been both celebrated for its potential and scrutinized for its challenges, including issues of interpretability, potential biases, and ethical concerns [14,15,16]. Local XAI methods, such as LORE, offer nuanced, instance-specific explanations that are essential for understanding complex AI decisions in medical contexts [17]. These explanations not only build confidence in AI technologies but also aid in identifying and correcting biases, ensuring ethical usage, and complying with healthcare regulations [18].

Interpretability at the local level allows healthcare professionals to comprehend the reasoning behind specific AI decisions, facilitating their trust in and collaboration with AI systems. Moreover, local explanations play a crucial role in elucidating AI outcomes to patients, empowering them with knowledge about their care processes and decisions.

**Challenges and Opportunities in Applying Local XAI**. While local XAI methods present a promising approach for enhancing interpretability in healthcare AI, they also introduce challenges such as maintaining model performance and ensuring the relevance and comprehensibility of explanations to end-users. Balancing the complexity of healthcare data with the need for understandable, actionable insights requires innovative solutions and continuous advancements in XAI techniques.

To address these challenges, this paper explores a variety of local XAI approaches, emphasizing the contribution of the LORE method for its ability to generate detailed, rule-based explanations relevant to individual cases. Such techniques are pivotal in translating the intricate patterns recognized by AI models into intelligible information, thereby improving the clinical utility of AI and fostering a collaborative healthcare environment.

**The Focused Approach of the XAI Project**. The XAI Project G.A. 834756) (https://xai-project.eu/, accessed on 14 March 2024) is an ERC-funded project focused entirely on the development of Explainable AI models. In the XAI Project, one of the most prominent domains of research has been local explainability in healthcare AI [19], aiming to refine and promote the application of methods like LORE for better decision-making processes. By concentrating on the development of local XAI techniques, the project seeks to address the specific interpretability needs of healthcare professionals and patients, ensuring that AI systems are not only accurate but also transparent and trustworthy.

In summary, this paper presents a focused narrative on the role of local XAI methods in healthcare, illustrating how such approaches can surmount the interpretability challenges posed by complex AI models. Through detailed case studies and analysis of the LORE method, it aims to showcase the tangible benefits of local explainability in improving patient care, ensuring ethical AI use, and enhancing the acceptance of AI technologies in medical settings.

## 2. Related Work

The exploration of Explainable Artificial Intelligence (XAI) in healthcare settings, particularly through the lens of local interpretation methods like Local Rule-Based Explanations (LORE) [1], has garnered considerable attention. This section delves into advancements in the interpretability and transparency of AI models in healthcare, emphasizing the importance of local XAI methods and their contributions to the field.

Recent efforts have aimed at enhancing the interpretability of healthcare AI models, with Rajkomar et al. proposing an “Explainable AI Framework for Health” that integrates rule-based models, gradient-based methods, and attention mechanisms for generating interpretable healthcare predictions [20]. This framework’s application to patient mortality prediction has enabled healthcare professionals to derive actionable insights from the model’s decision-making process, showcasing the utility of comprehensive XAI approaches in clinical settings.

Ribeiro et al.’s Anchors method represents another noteworthy advancement, offering rule-based explanations tailored to individual predictions [21]. By concentrating on locally faithful explanations, the Anchors method has empowered healthcare practitioners with a clearer understanding of the factors influencing AI predictions in specific scenarios. Its application across various healthcare domains underscores the method’s effectiveness in improving interpretability at the local level.

The SHAP (SHapley Additive exPlanations) framework, introduced by Lundberg et al., utilizes game theory to allocate feature importance values for individual predictions, thus providing a detailed view of how each input affects the model’s output [22]. Applied in contexts such as hospital readmission prediction, disease progression modeling, and electronic health records analysis, SHAP has been instrumental in enhancing transparency and interpretability in healthcare AI.

Furthermore, interpretability techniques like LIME (Local Interpretable Model-agnostic Explanations) and LORE (Local Rule-Based Explanations) have seen wide adoption in the healthcare sector [1,23]. LIME’s approach to generating local explanations by approximating complex model decision boundaries complements LORE’s use of a genetic algorithm to create a synthetic neighborhood for a local interpretable predictor. This predictor, in turn, facilitates the generation of meaningful explanations that include decision rules and counterfactual scenarios, thereby illuminating the influence of specific factors on outcomes [1]. These techniques have been applied to a range of healthcare domains, from disease prediction to medical imaging and clinical decision support, demonstrating their versatility and impact.

Caruana et al.’s work on developing intelligible models for healthcare contexts, such as pneumonia risk prediction and hospital readmission, further highlights the progress in creating interpretable AI systems for clinical use [24]. By employing decision trees and rule-based models, their research has contributed significantly to the field, enhancing both the transparency and the adoption of AI in clinical practice.

The integration of domain knowledge and expert input into XAI approaches marks an evolving research direction, promising to enrich interpretability and align AI decision-making processes with established medical practices. This blend of technical innovation and domain expertise is crucial for advancing the application of local XAI methods in healthcare, ensuring that AI-assisted decision making is both transparent and grounded in clinical realities [25].

In conclusion, advancements in XAI for healthcare, particularly the focus on local methods, highlight a growing commitment to enhancing AI model transparency and interpretability. These efforts underscore the field’s progress towards developing AI systems that are not only technically proficient but also understandable and trustworthy for healthcare professionals and patients alike, fostering improved decision making, patient care, and adherence to regulatory standards.

## 3. Methodology

Before discussing the specific methodologies supporting our research, it is crucial to contextualize our work within the broader landscape of artificial intelligence technologies, particularly deep learning. Deep learning [26], a subset of machine learning, has emerged as a transformative force in various domains, including healthcare. It refers to the development of algorithms that can learn and make decisions or predictions based on data. These algorithms, known as neural networks, are designed to mimic the human brain’s architecture and function, processing vast numbers of data to identify patterns and insights that are not immediately apparent to human observers.

As outlined in the seminal work [27], deep learning involves multi-layered neural networks that learn and make inferences from data in a way that captures the complexity and subtlety of the information being processed. This capability makes deep learning particularly valuable in healthcare, where the ability to analyze and interpret complex medical data can lead to more accurate diagnoses, personalized treatment plans, and, ultimately, better patient outcomes.

The key methodologies used in the project are related to the LORE method, introduced by Guidotti et al. [1]. LORE is a powerful framework for generating local and interpretable explanations for machine learning models. LORE utilizes a genetic algorithm to create a synthetic neighborhood, which serves as the basis for training a local interpretable predictor. This predictor captures the underlying logic of the model’s decision-making process, enabling the derivation of meaningful explanations.

One of the key characteristics of LORE is its ability to provide transparent and understandable explanations for individual predictions. By focusing on local interpretability, LORE aims to explain the reasoning behind a specific prediction rather than the overall behavior of the model. This makes it particularly useful in situations where interpretability at the instance level is crucial, such as in healthcare and finance.

The explanations consist of two main components. First, a decision rule is derived from the logic of the local interpretable predictor. This decision rule sheds light on the factors that influenced the model’s decision, providing insights into the important features and their corresponding weights. This information helps in understanding the key drivers behind the prediction. Additionally, LORE produces a set of counterfactual rules as part of the explanation. These counterfactual rules suggest modifications to the instance’s features that would lead to a different outcome. By providing actionable suggestions for changing the input variables, LORE enables users to explore what-if scenarios and understand how small changes can influence the model’s predictions.

The availability of the LORE framework, along with the accompanying code (https://github.com/riccotti/LORE, accessed on 14 March 2024), facilitates its adoption and implementation in various domains. In the next sections, different research projects are described. They leverage over the LORE methodology from different points of view.

### Detailed LORE Framework

LORE operates on the principle of providing instance-specific explanations by creating a local, interpretable model around a prediction of interest. It begins by selecting an instance for which an explanation is desired. Then, it generates a synthetic dataset that mimics the locality of the original instance through a genetic algorithm. This local dataset is used to train a simple, interpretable model, such as a decision tree, which serves to approximate the behavior of the complex model near the instance. The explanation is then derived from this interpretable model in the form of rules, which highlight the decision-making process for the specific instance.

**Implementation Steps**: Selection of Target Instance: Choose the specific prediction or instance that requires explanation.

Synthetic Neighborhood Generation: Utilize a genetic algorithm to generate a synthetic dataset that represents the local decision boundary around the target instance.

Training of Interpretable Model: Train a simple model, like a decision tree, on this synthetic dataset to capture the local decision logic of the complex model.

Derivation of Explanation: Extract rules from the interpretable model that explain the prediction of the target instance. These rules offer insights into which features and conditions influence the decision.

LORE’s ability to provide clear, case-specific explanations makes it highly valuable in healthcare settings, where understanding the rationale behind AI-driven diagnostic or prognostic predictions is crucial. For instance, LORE has been applied to interpret AI decisions in predicting patient outcomes, understanding disease progression, and personalizing treatment plans. Its interpretability supports clinical decision making, enhances trust among medical practitioners, and facilitates patient communication.

LORE distinguishes itself from other XAI methods like LIME or SHAP primarily through its emphasis on generating a synthetic neighborhood around an instance. This approach allows LORE to provide highly localized explanations that are directly relevant to the specific case at hand. While LIME also focuses on local interpretability, it approximates the model’s decision boundary linearly, which might not capture complex nonlinear relationships as effectively as LORE’s method. SHAP, on the other hand, provides a global interpretation by assigning importance values to features based on their contribution to the model’s output. LORE’s advantage lies in its detailed, rule-based explanations that can be more intuitively understood by healthcare professionals for specific patient cases.

## 4. XAI Frameworks for Healthcare

### 4.1. DoctorXAI

An ontology-based approach, as described in “Doctor XAI: an ontology-based approach to black-box sequential data classification explanations” [28], aims to provide explanations for the black-box predicting of multi-labeled, sequential, ontology-linked data [29]. The methodology involves the use of ontologies, which are formal representations of knowledge, to capture domain-specific concepts and relationships [30]. This paper focuses on explaining Doctor AI [31], a multi-label classifier which takes as input the clinical history of a patient in order to predict the next visit.

In greater detail, the methodology begins by selecting real neighbors, which are data points closest to the instance to be explained, either through a standard distance metric or ontology-based similarities. A synthetic neighborhood is then generated by perturbing the real neighbors to maintain locality. The challenge lies in generating meaningful synthetic instances, and here the authors leverage the ICD-9 ontology to ensure the expressiveness of the neighborhood. Unlike other techniques, the perturbations are not applied directly to the instance to be explained to prevent homogeneity in the neighborhood. Two alternative paths are followed; see Figure 1: the red path involves normal perturbation and encoding/decoding steps to transform the data for interpretable models, while the blue path involves ontological perturbation directly on sequential data. In both paths, the synthetic neighborhood is labeled by the black-box model and used to train an interpretable model, such as a multi-label decision tree. Rule-based explanations are then extracted from the decision tree. The methodology extends the general framework with novel contributions for dealing with structured and sequential data. These components can be independently incorporated into the explanation pipeline based on the nature of the data point to be explained.

The authors use the MIMIC-III dataset [32] which contains de-identified health-related data associated with over 40,000 patients who stayed in critical care units of the Beth Israel Deaconess Medical Center between 2001 and 2012. In the experiments conducted, the ontology-based approach is compared against other existing methods for generating explanations. Various evaluation metrics, such as accuracy, coverage, and coherence, may be employed to assess the quality and comprehensibility of the explanations provided by the approach. The results of the experiments highlight the efficacy of the ontology-based approach in generating meaningful explanations for black-box sequential data classification models. The approach demonstrates its ability to capture domain-specific knowledge, extract relevant features, and provide interpretable explanations that enhance the understanding of the underlying decision-making process (DoctorXAI demo: https://kdd.isti.cnr.it/DrXAI-viz/, accessed on 14 March 2024); see Figure 2.

This paper suggests future work in exploring alternative synthetic neighbor generation for sequential data and assessing the impact of random components. Additionally, the authors plan to extend the technique to explain how black-box regressors predict continuous outcomes, which is relevant in healthcare for risk stratification prediction tasks.

DoctorXAI has been proven to enhance physicians’ interactions with machine learning models. In the work of Panigutti et al. [33], the authors conduct a rigorous, survey-based analysis of physicians’ interactions with an AI-based Clinical Decision Support System equipped with DoctorXAI. The results indicate that the explanations provided by DoctorXAI enhance the trust between physicians and the AI-based system.

### 4.2. FairLens

The pervasive application of AI in critical areas, especially healthcare, has brought to the forefront the challenges associated with unintended biases. These biases, if unchecked, can have profound implications, especially when decisions impact patient care. Recognizing this, in [34], the authors present FairLens, a tool designed to audit, discover, and explain biases in AI systems, particularly those deployed in clinical settings. A general overview of FailLens is presented in Figure 3.

FairLens is rooted in a multi-step approach:Stratification of Patient Data: Before any analysis, the tool stratifies available patient data based on various attributes, including age, ethnicity, gender, and insurance type. This stratification allows for a more granular analysis of how the AI model performs across different patient subgroups.Performance Assessment: Once stratified, FairLens evaluates the model’s performance on these subgroups. It identifies areas where the model might be underperforming or showing biases. This step is crucial as it pinpoints specific patient groups that might be adversely affected by the model’s decisions.Explanation of Model Errors: Going beyond mere identification, FairLens delves into the reasons behind the model’s errors. Using advanced XAI techniques, the tool determines which elements in a patient’s clinical history contribute to the model’s inaccuracies. This step is pivotal as it not only highlights the errors but also provides insights into why they occur.

The FairLens pipeline is a systematic process that ensures a comprehensive audit of the AI model. At first, the patient data are divided based on predefined conditions, creating various groups. Each group is then scored based on the model’s performance, providing a quantitative measure of the model’s accuracy for that subgroup.

Post-scoring, the groups are ranked. This ranking serves as an indicator, highlighting groups where the model’s performance is low-grade. Selected groups (based on ranking or expert input) undergo a detailed inspection. Here, the model’s predictions are compared against actual data to identify over-represented or under-represented conditions.

For mislabeled conditions, FairLens provides explanations. It identifies clinical conditions that are frequently misclassified and elucidates the elements in patients’ histories that influence these misclassifications. The entire analysis culminates in a comprehensive report that details the findings, providing both a bird’s-eye view and in-depth insights.

FairLens represents a significant step forward in ensuring that AI models, especially those in healthcare, are free from detrimental biases. By providing a systematic methodology and a clear pipeline, it offers a robust framework for auditing black-box clinical decision support systems. The tool’s ability to not just identify but also explain biases makes it invaluable for healthcare professionals, ensuring that AI-driven decisions are both accurate and fair.

### 4.3. MARLENA

Machine learning models, especially deep learning ones, have become central to many decision-making systems in healthcare. They assist in diagnosis, predict disease spread, and help in identifying high-risk patient groups. However, the inherent lack of transparency in these models can lead to mistrust, potential biases, and even legal implications. MARLENA (Multi-label Rule-based ExplaNAtions) [35] is introduced as a solution to the interpretability challenge. It is designed to provide explanations for decisions made by multi-label black-box classifiers. The main idea of miming the local behavior of a black-box is common with other approaches such as LIME [23] and LORE [1]. However, none of these approaches is applicable to explain multi-label black-box classifiers. An overview of MARLENA is presented in Figure 4. The novel methodology is broken down into three primary steps:Synthetic Neighborhood Generation: Before explaining a decision, MARLENA first creates a synthetic neighborhood around the instance in question. This neighborhood is populated with data points that are similar to the instance, ensuring that the explanation is localized and relevant.Learning a Decision Tree: Using the synthetic neighborhood, MARLENA constructs a decision tree. Decision trees are inherently interpretable, making them suitable for this purpose.Deriving Decision Rules: From the constructed decision tree, MARLENA extracts decision rules that provide a clear and concise explanation for the black-box decision concerning the instance.

The core methodology revolves around the generation of a neighborhood around the instance that needs elucidation. This is crucial because the explanation is intended to be local, focusing on the behavior of the black-box classifier concerning that specific instance.

To generate this neighborhood, MARLENA employs two strategies. **Constructing a Core Real Neighborhood**: This involves identifying real instances from the dataset that are close to the instance in both the feature space and decision space. This real neighborhood provides a foundation upon which synthetic neighbors can be generated. **Generating Synthetic Neighbors**: Based on the empirical distributions of the instance’s features derived from the real neighborhood, MARLENA generates synthetic neighbors. These neighbors are designed to mimic the behavior of the black-box classifier in the vicinity of the instance. Once the neighborhood is established, MARLENA proceeds with the construction of the decision tree and the extraction of decision rules.

MARLENA offers a structured approach to demystifying decisions made by multi-label black-box classifiers. By focusing on local explanations and leveraging interpretable models like decision trees, the method ensures that the explanations provided are both meaningful and relevant.

### 4.4. The International Skin Imaging Collaboration

ABELE (Adversarial Black-box Explainer generating Latent Exemplars) [36] is a local model-agnostic explainer that takes an image and a black-box classifier as input and returns a set of exemplar and counter-exemplar images, as well as a saliency map.

Exemplars and counter-exemplars are synthetically generated images classified with the same outcome as the input image and with an outcome other than the input image, respectively. They can be visually analyzed to understand the reasons for the decision. The saliency map highlights the areas of the input image that contribute to its classification and areas that push it into another class.

ABELE works by generating a neighborhood in the latent feature space using an Adversarial Autoencoder (AAE [37]). The image to be explained is passed as input to the AAE where the encoder returns the latent representation using latent features. A genetic approach maximizing a fitness function was adopted to accomplish the neighborhood generation. In this respect, ABELE takes advantage of a latent version of LORE.

After the generation process, for any instance in the neighborhood, ABELE checks the validity of the instance by querying the discriminator and decoding it into an image. Then, it queries the black-box classifier with the image to obtain the class. Given the local neighborhood, ABELE builds a decision tree classifier trained on the neighborhood labeled with the black-box classifier. The surrogate tree is intended to locally mimic the behavior of the black-box classifier in the neighborhood. It extracts the decision rule and counter-factual rules enabling the generation of exemplars and counter-exemplars.

The overall effectiveness of ABELE lies in the goodness of the encoder and decoder function adopted. The better the AAE, the more realistic and useful the explanations will be.

In recent years, deep learning, particularly through convolutional neural networks (CNNs), has significantly advanced the detection and diagnosis of skin cancer lesions [38,39,40,41], achieving diagnostic accuracies comparable to dermatologists. This progress promises improved early detection rates and broader access to high-quality diagnostic services. However, the effectiveness of these models in clinical settings hinges on their interpretability and the transparency of their decision making, ensuring healthcare professionals can integrate AI insights confidently into patient care. In [42,43,44,45], a case study on skin lesion diagnosis using a ResNet classifier trained on the ISIC (International Skin Imaging Collaboration) dataset is presented. The classifier’s decisions are explained using ABELE.

A user interaction module was implemented as a web application to present the results of the classification and the corresponding explanation. The module communicates with a backend that exposes the functionalities of the black-box and ABELE via a RESTful interface (https://kdd.isti.cnr.it/isic_viz/, accessed on 14 March 2024). The visual space of the application is organized into two sections (see Figure 5). The upper part shows the instance under analysis with the classification returned by the ResNet on the left and a synthetic counter-exemplar image returned by ABELE on the right. The lower part of the module shows four exemplars, i.e., a set of images returned by ABELE that have the same label assigned by the ResNet to the instance under analysis.

The customization of the autoencoder, specifically an Adversarial Autoencoder (AAE), is crucial in this case study due to the complexity of the image classification task and the limitations of the dataset. The ISIC dataset, which is used for training the ResNet classifier [46], presents challenges such as fragmentation, imbalance, lack of uniform digitization, and shortage of data. Training an AAE in a standard fashion without addressing these issues results in poor performance, mainly due to a persistent mode collapse.

To overcome these challenges, a collection of cutting-edge techniques were implemented, including Mini Batch Discrimination and Denoising autoencoders. The model of AAE adopted is a Progressive Growing AAE, which helps achieve more stable training of generative models for high-resolution images. The main idea is to start with a very low-resolution image and, step by step, add blocks of layers that simultaneously increase the output size of the generator model and the input size of the discriminator model until the desired size is achieved. In this case, the desired size is 224 × 224 pixels.

The latent space dimension is kept fixed, so the discriminator always takes as input tensors of the same size. The incremental addition of the layers allows the Progressive Growing AAE to first learn large-scale structure and progressively shift the attention to finer detail. This approach greatly reduces mode collapse and enables the generation of varied and high-quality synthetic skin lesion images.

The customization of the AAE is necessary to make it usable for the complex image classification task addressed by the ResNet classifier. After a thorough fine-tuning of all three network structures (encoder, decoder, and discriminator), the Progressive Growing AAE with 256 latent features achieves a reconstruction error measure through RMSE that ranges from 0.08 to 0.24 depending on whether the most common or the rarest skin lesion class is considered. This customization allows ABELE to generate meaningful explanations and can be tested in a survey involving real participants.

A survey was conducted involving domain experts, beginners, and unskilled people to assess the effectiveness of the explanations provided by ABELE. The results of the survey show that the usage of explanations increases trust and confidence in the automatic decision system. This phenomenon is more evident among domain experts and people with the highest level of education. After receiving wrong advice from an AI model, domain experts tend to decrease their trust in the same model for future analysis.

The survey was designed to validate the effectiveness of the explanations returned by ABELE for skin lesion diagnosis. The main purpose was to validate the effectiveness of the explanations in assisting doctors and medical experts in the diagnosis and treatment of skin cancers, as well as to investigate their confidence in automatic diagnosis models based on black-boxes and on the explanations provided by the explainer.

The survey was organized into ten questions composed of various points. Participants were presented with an unlabeled skin lesion image randomly chosen from the dataset and its explanation as generated by ABELE. They were asked to classify the given image among two different given classes exploiting the explanation. Participants were also presented with a labeled image and they were asked to quantify their level of confidence in the black-box classification. The same labeled image was then presented with the visual aid of the explanation returned by ABELE, and they were asked to quantify their confidence once more after looking at the explanations. Participants were also asked to quantify how much the exemplars and counter-exemplars helped them to classify skin lesion images in accordance with the AI and how much they trust the explanations.

The survey results support the hypothesis that explanation methods without a consistent validation are not useful. The results also highlight the analysis of the latent space of the autoencoder made available by ABELE. The latent space analysis suggests an interesting separation of the images that can hopefully be helpful in separating similar classes of skin lesions that are frequently misclassified by humans.

## 5. Discussion and Conclusions

This paper has discussed the integration and application of Explainable Artificial Intelligence (XAI) within healthcare, focusing on the challenges, developments, and future directions of XAI in medical diagnostics and patient care. Throughout the ERC XAI project, significant progress has been made in understanding the complex dynamics of XAI in healthcare, its potential benefits, and the inherent challenges encountered.

**Challenges and Overcoming Strategies**. One of the primary challenges encountered during the project was the complexity of medical data and the difficulty of generating accurate, comprehensible explanations for AI-based decisions. The heterogeneity of healthcare data, along with the high stakes involved in medical decision making, necessitates explanations that are not only technically accurate but also easily understandable by healthcare professionals. To address this, we adopted a multi-faceted approach integrating local explanation generation with formal verification methods. This approach ensured that explanations were both locally relevant and globally consistent with the classifier’s logic, thereby enhancing the trustworthiness and explainability of AI systems in healthcare.

**Importance and Challenges in Healthcare**. In the medical field, the adoption of XAI methods faces unique challenges, including ensuring patient privacy, dealing with high-dimensional data, and the critical need for accuracy. Despite these challenges, the importance of XAI in healthcare cannot be overstated. Detailed, understandable AI explanations empower clinicians to make informed decisions, foster patient trust, and enhance the overall effectiveness of medical treatments. Our work, through projects like DoctorXAI and MARLENA, demonstrates the feasibility and value of applying XAI to a range of healthcare applications, from diagnosing skin lesions to evaluating cardiac risk.

**Future Directions and Methodologies**. Looking forward, the field of XAI in healthcare is poised for rapid growth and innovation. Future methodologies should focus on improving the robustness and versatility of explanation models, incorporating more diverse data types (e.g., genomic data, electronic health records), and exploring new forms of explanations (e.g., visual explanations, interactive models). Additionally, there is a need for more interdisciplinary research that combines insights from data science, medicine, psychology, and ethics to develop XAI systems that are not only technically proficient but also ethically sound and aligned with patient care goals.

**Integrating XAI with Medicine**. The future of XAI in healthcare lies in its seamless integration as a decision support system, complementing, not replacing, human expertise. For this to be realized, XAI systems must be designed with openness, transparency, and interpretability at their core. This approach will ensure that healthcare professionals can trust and effectively use AI recommendations, leading to improved patient outcomes.

**Building Trust among Healthcare Professionals**. To build and maintain trust in XAI systems among medical practitioners, it is essential to focus on user-centered design principles, ensuring that explanations are relevant, actionable, and tailored to the user’s expertise level. Avoiding overly complex or opaque AI models and instead emphasizing the transparency and reliability of explanations will be key. Additionally, ongoing education and training for healthcare professionals on the capabilities and limitations of AI will play a critical role in fostering a collaborative environment where AI and human expertise work hand in hand.

In our exploration of Explainable Artificial Intelligence within healthcare, a significant aspect that emerges is the imperative of human–machine collaborative decision making. This symbiotic interaction underscores the philosophy that the greater involvement of human judgment alongside AI can significantly enhance the explainability and ethical dimensions of healthcare decisions. The interplay between human insight and AI’s analytical skills promises to elevate clinical decision making to new heights, fostering a deeper trust and understanding between healthcare providers and the technology they leverage. Moreover, this collaboration can serve as a cornerstone for ethical AI use, ensuring decisions are not only accurate but also transparent and aligned with patient values and needs. As we look towards the future, the integration of human expertise with sophisticated AI algorithms will be crucial in navigating the complex ethical landscape of healthcare, ensuring that AI-assisted decisions are made with a comprehensive understanding of patient care, thereby reinforcing the essence of medicine and underscoring its profound human-centric nature.

In conclusion, the integration of XAI into healthcare holds tremendous promise for enhancing medical diagnostics, patient care, and treatment outcomes. By continuing to address the challenges, leveraging the strengths of AI and human expertise, and focusing on patient-centered outcomes, the future of XAI in healthcare is bright. With sustained research and development, XAI can become an indispensable tool in the medical field, offering insights and explanations that support clinical decision making and contribute to the advancement of personalized medicine.

## 6. Future and Ongoing Work

As we venture into the future of XAI in healthcare, the emphasis on human–machine collaborative decision making will play a pivotal role in shaping research and development directions. Our forthcoming projects aim to delve deeper into models and frameworks that not only advance the technical capabilities of XAI but also enhance its alignment with human expertise and ethical considerations in clinical settings. This will involve developing systems that are capable of incorporating feedback from healthcare professionals directly into the AI learning process, thus refining the accuracy and applicability of AI outputs in a real-world context. Moreover, the exploration of the ethical and regulatory implications of these collaborative systems will be fundamental. By fostering a more integrated approach to AI in healthcare, where technology and human expertise complement each other, we anticipate not only bridging the gap between AI’s potential and its practical application but also contributing to the development of AI systems that are both ethically responsible and highly effective in enhancing patient care outcomes.

We present a sketch of our ongoing projects that aim to further develop and expand this promising area.

### 6.1. Cardiac Risk Evaluator

In an upcoming work we present VERIFAI-LORE, a framework designed to enhance the trustworthiness and explainability of AI-based classifiers. This is achieved through a unique integration of search-based approaches, machine-learned explanations, satisfiability solving, and theorem proving. Specifically, it utilizes LORE (Local Rule-Based Explanations) to generate local explanations for classifications by sampling around an instance and constructing a decision tree. These explanations comprise logic rules and counter-rules, indicating the attributes that contributed to a classification and conditions for a different classification, respectively.

However, recognizing that these explanations may be locally valid but underconstrained for certain instances, VERIFAI-LORE introduces a formal verification step. This step involves translating the model into Java and writing a JML (Java Modeling Language) contract in the form of precondition–postcondition pairs to verify the consistency of an explanation with the classifier through theorem proving. This allows the framework to ensure that explanations are not only statistically valid but also logically consistent across all possible inputs, addressing the challenge of underconstrained explanations.

The integration of explanation generation and formal verification in the VERIFAI-LORE framework aims to provide globally consistent and locally valid explanations for each classification, thereby enhancing classifier trustworthiness. The framework is evaluated in a case study on the prognosis of Acute Coronary Syndrome (ACS) [47], demonstrating its capability to provide classifications with associated confidence levels and explanations that are formally verified for consistency. The evaluation shows that checking the JML contract for explanations takes on average 12.6 s and 304.5 MB for consistent explanations and 68.7 s and 326.5 MB for underconstrained explanations, indicating that underconstrained explanations occur infrequently. This contributes to advancing trustworthy and explainable classification. A visual interface (https://kdd.isti.cnr.it/cre_vue/#/, accessed on 14 March 2024) has been made public and allows for the appreciation of the quality of explanations based on custom inputs entered by the user. Figure 6 demonstrates the explanation of the outcome related to test data.

### 6.2. Prostate Imaging Cancer AI

This ongoing project involves the application of local explanation algorithms to a different context—a prostate cancer MRI dataset [48,49,50,51]. This dataset, collected in collaboration with the Prostate Cancer Unit at Ospedale Careggi of Florence and the PI-CAI Grand Challenge (https://pi-cai.grand-challenge.org/, accessed on 14 March 2024), is composed of T2-weighted, Apparent Diffusion Coefficient (ADC), and DWI magnetic resonance images. Our goal is to harness the power of local methods in generating meaningful explanations for complex imaging analyses to improve our understanding of prostate cancer diagnostics (Figure 7). The project will not only focus on enhancing the explainability of local methods but will also delve into an innovative realm of cross-domain explanations between different modalities, i.e., image and tabular data. By doing so, we plan to bridge the gap between different imaging modalities and foster a more integrated, comprehensive understanding of prostate cancer diagnosis, thereby contributing to more effective patient management and treatment outcomes.

### 6.3. Diabetology

An upcoming project in collaboration with the Diabetology Department at Cisanello University Hospital (Pisa, Italy) explores the application of Explainable AI data describing patients’ conditions before and after liver transplantation. Leveraging a fairly small dataset of roughly 450 patients including pre- and post-operative factors, we aim to develop Explainable AI models with the objective of answering a number of questions regarding the effect of liver transplantation on patients [52]. Using explanations as a lens to inspect the models, we are going to investigate possible effects on the onset of diabetes, patient survival, and graft survival. The key challenges involve the highly imbalanced dataset, with the survival rate significantly higher than mortality, and the low number of samples. We plan to apply data balancing techniques cautiously to maintain the integrity of real-world examples so that Explainable AI techniques can still extract meaningful insights from the predictive models. The ultimate goal is to identify key factors contributing to survival rates and understand the relationship between a patient’s diabetic condition and survival outcome, thereby offering physicians a way to find new and actionable insights.

### 6.4. DoctorXAI++

DoctorXAI [28] has been proven to be highly beneficial to clinicians for understanding a machine learning model’s decision and for improving trust in the model output [33]. The various components of the DoctorXAI architecture, however, can be improved, in light of recent advancements in the model’s performance and in generative AI techniques. This project has the objective of improving DoctorXAI by applying more modern deep learning models to the updated MIMICIV [53] dataset with the ICD10 ontology. The initial experiments we have performed in this direction have the objective of investigating the synthetic neighborhood generation mechanism of DoctorXAI, improving it with the use of Large Language Models (LLMs), trained to approximate the original data distribution. LLMs show the potential of learning the original ontology directly from the data. This may indicate that LLMs can be used both to perform predictions and to replicate ontology-based explanations using only the raw ICD data. A general schema of our approach is shown in Figure 8.

## Figures and Tables

**Figure 1 bioengineering-11-00369-f001:**
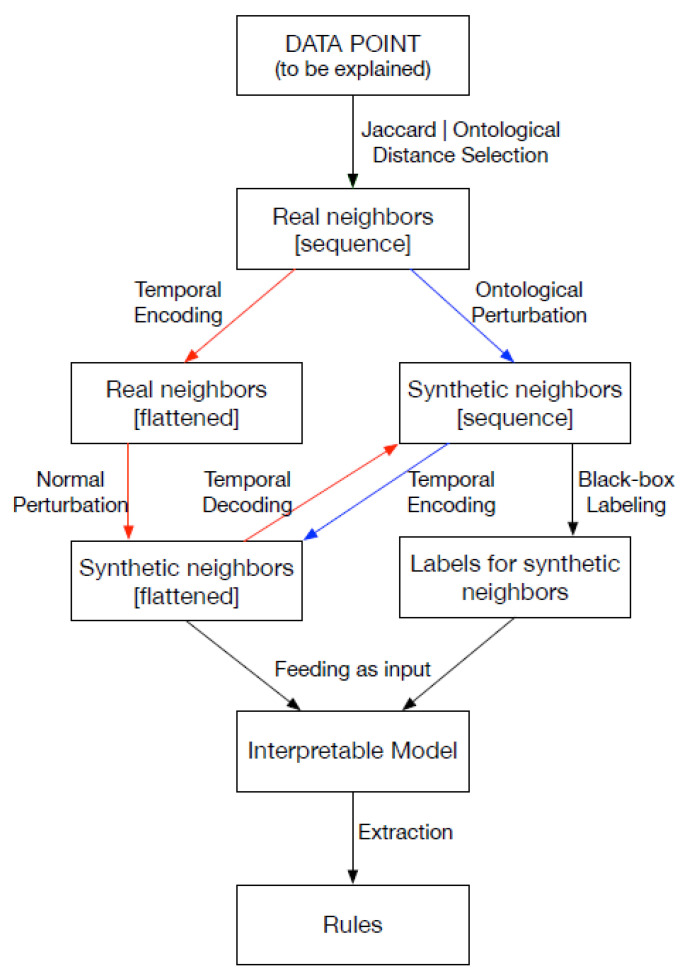
DoctorXAI explanation pipeline from [28].

**Figure 2 bioengineering-11-00369-f002:**
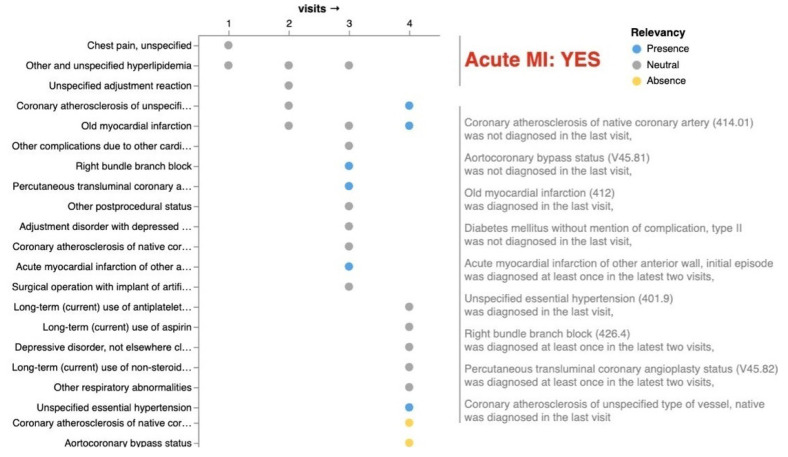
Doctor XAI explanation presented to the participants of the experiment. In the image, distinct dots represent a single visit of a patient, and distinct colors represent the relevance of each dot to the algorithmic decision. Dots associated with irrelevant conditions remain gray, whereas those deemed relevant are depicted in blue. Additionally, DoctorXAI highlights, as yellow dots, any conditions absent from the patient’s clinical history that could have altered the algorithmic suggestion.

**Figure 3 bioengineering-11-00369-f003:**
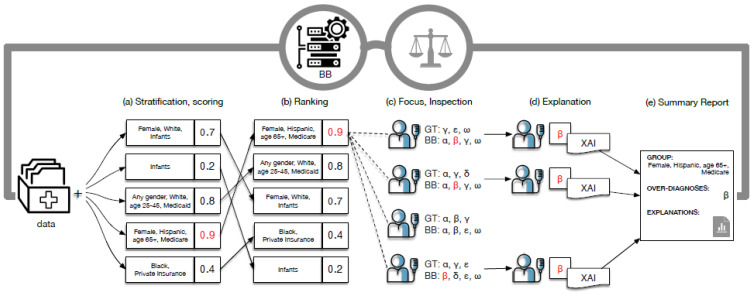
FairLens pipeline from [34].

**Figure 4 bioengineering-11-00369-f004:**
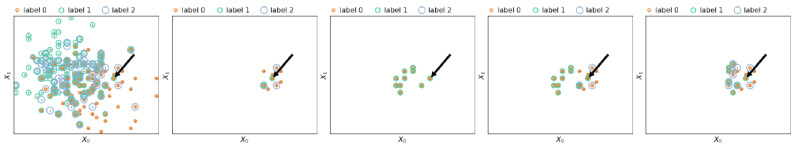
From [35]. A graphical representation of mixed neighborhood generation starting from a sample dataset with three different labels. (1st) a dataset sample, and the arrow points out the instance to explain x; mixed neighborhood generation: (2nd) real instances close to x with respect to the feature space; (3rd) real instances close to x with respect to the target space; (4th) a merge of the previous sets of instances. Unified core real neighborhood: (5th) real instances close to x with respect to feature and target spaces, i.e., the real core neighborhood.

**Figure 5 bioengineering-11-00369-f005:**
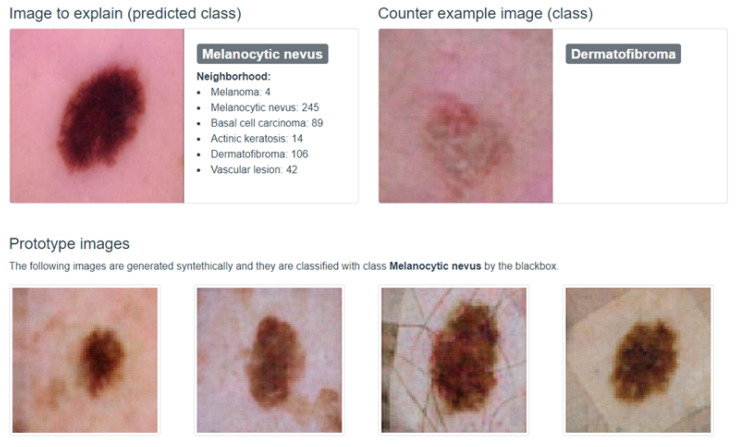
A user visualization module to present the classification and the corresponding explanation. The upper part presents the input instance and a counter-exemplar. The lower part shows four exemplars that share the same class as the input.

**Figure 6 bioengineering-11-00369-f006:**
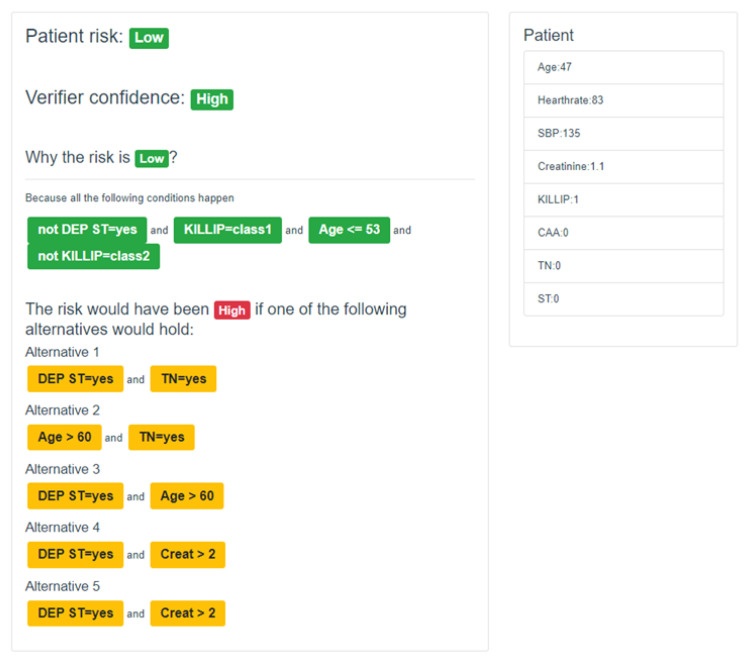
User visualization module to present the classification and the corresponding explanation of the outcome related to test data consisting of *age: 47, heart rate: 83, Systolic Blood Pressure: 135, creatinine: 1.1, cardiac arrest at admission: no, ST segment deviation on EKG: no, TN (abnormal cardiac enzymes): no, and KILLIP class: no CHF*.

**Figure 7 bioengineering-11-00369-f007:**
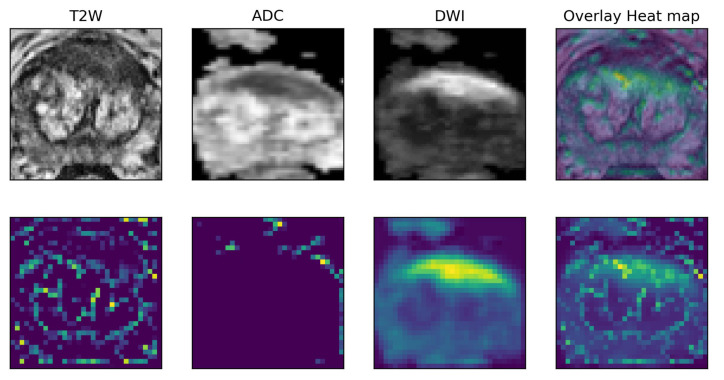
A saliency map extracted from a 3-channel multimodal classifier comprising T2W, ADC, and DWI images of a prostatic gland with a high-grade prostate lesion. Each activation map sizes the contribution of each modality.

**Figure 8 bioengineering-11-00369-f008:**
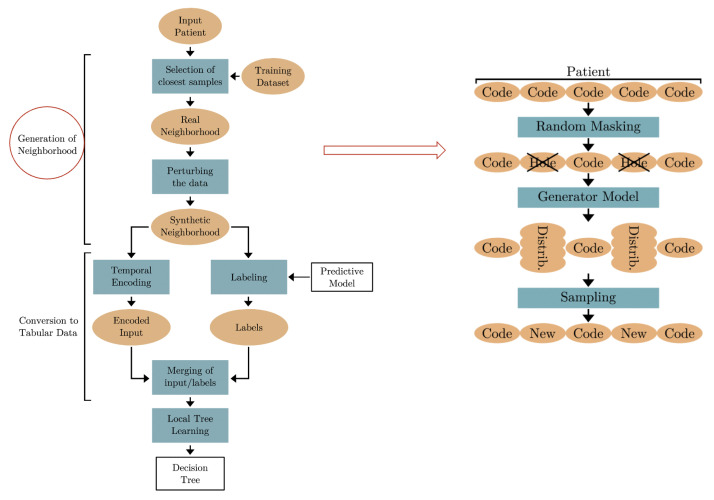
General schema of DoctorXAI++.

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
