# Peer review of "Towards Transparent Healthcare: Advancing Local Explanation Methods in Explainable Artificial Intelligence"

_bioengineering, 2024, doi:10.3390/bioengineering11040369_

Round 1

Reviewer 1 Report

Comments and Suggestions for Authors

Comment 1:

I wonder the adequacy of the concept of “local” explainable .... It sounds to me the concept of “specific” may be more adequate. The concept of specific is to the concept of general.

Comment 2:

This paper seems to review the methodologies on local XAI. It needs to include the definition of deep learning, the latest comprehensive one is for example ‘On Definition of Deep Learning, 2018 World Automation Congress (WAC), Stevenson, WA, USA, 2018, pp. 1-5, doi: 10.23919/WAC.2018.8430387’. In that paper, a novel methodology for making a transparent deep learning, though not call it XAI, is outlined, which is based on the general knowledge architecture called FCBPSS (on domain modeling .....). The novel methodology has a promise to derive physics laws, etc.

Comment 3:

I am a little bit confused by the nature of this paper along with its objective. Section 4 is “Project”. This heading confused me, wondering if the paper is about project report or about a proposal.

Comment 4:

How about the human-machine collaborative decision making process, which may be interpreted as a kind of XAI. Philosophically, more involvements of humans may lead to more explainable.

Comments on the Quality of English Language

English is good.

Author Response

Dear Editor and Reviewers:
Thank you for the effort in reviewing our manuscript, Towards Transparent Healthcare: Advancing Local Explanation Methods in XAI. In the attached file we address the concerns of the reviewers, hoping that our efforts can clarify and improve our contribution to a level that you deem acceptable.
Yours sincerely,

The Authors

Reviewer 2 Report

Comments and Suggestions for Authors

This study introduces a method that focuses on the use of locally interpretable artificial intelligence (XAI) in healthcare and healthcare environments, particularly local rule-based interpretation (LORE) technology.However, there are still several problems that need to be improved.

1.  Please provide a detailed description of the Local Rule Based Explanations (LORE) technology used, including its working principle, implementation steps, and specific applications in the medical field. Discuss the comparison of this technology with other interpretive artificial intelligence methods and explain its advantages.

2. The author provides detailed technical background and application examples on the use of local interpretation methods such as LORE and their specific role in improving the understanding of machine learning models by doctors and patients. But in order to help non professional readers better understand the practical application of these methods, it is recommended to add more example analysis and visual display, while clarifying the advantages and disadvantages of different interpretation methods.

3. The paper mentions multiple research projects and conference papers, such as the DoctorXai++architecture, FairLens method for auditing black box clinical decision support systems, interpretation work for multi label classifiers, and examples and counterexamples of skin lesion classifiers. However, the article did not provide a detailed introduction to the specific experimental design and quantitative evaluation indicators of the experimental results of these studies. In order to enhance the scientific rigor of the article, statistical verification of these experimental details and results should be supplemented.

4. Lack of specific experimental design and result analysis. Please provide additional experimental data, including the dataset used, experimental settings, evaluation metrics, and obtained results. And conduct in-depth discussions on the experimental results, analyze the specific performance of LORE technology in improving the effectiveness, fairness, and compliance with regulatory standards in medical decision-making.

Comments on the Quality of English Language

This study introduces a method that focuses on the use of locally interpretable artificial intelligence (XAI) in healthcare and healthcare environments, particularly local rule-based interpretation (LORE) technology.However, there are still several problems that need to be improved.

1.  Please provide a detailed description of the Local Rule Based Explanations (LORE) technology used, including its working principle, implementation steps, and specific applications in the medical field. Discuss the comparison of this technology with other interpretive artificial intelligence methods and explain its advantages.

2. The author provides detailed technical background and application examples on the use of local interpretation methods such as LORE and their specific role in improving the understanding of machine learning models by doctors and patients. But in order to help non professional readers better understand the practical application of these methods, it is recommended to add more example analysis and visual display, while clarifying the advantages and disadvantages of different interpretation methods.

3. The paper mentions multiple research projects and conference papers, such as the DoctorXai++architecture, FairLens method for auditing black box clinical decision support systems, interpretation work for multi label classifiers, and examples and counterexamples of skin lesion classifiers. However, the article did not provide a detailed introduction to the specific experimental design and quantitative evaluation indicators of the experimental results of these studies. In order to enhance the scientific rigor of the article, statistical verification of these experimental details and results should be supplemented.

4. Lack of specific experimental design and result analysis. Please provide additional experimental data, including the dataset used, experimental settings, evaluation metrics, and obtained results. And conduct in-depth discussions on the experimental results, analyze the specific performance of LORE technology in improving the effectiveness, fairness, and compliance with regulatory standards in medical decision-making.

Author Response

(The authors gave the same response as above.)

Round 2

Reviewer 1 Report

Comments and Suggestions for Authors

I am satisfied the authors' revised manuscript along with their rebuttal.

Comments on the Quality of English Language

Good.

Author Response

Thank you for you comments and suggestions. They have improved the quality of the contribution.

The authors.